# On Generating Abstract Explanations via Knowledge Forgetting

## Abstract

In this paper, we investigate the problem of generating explanations from the context of Human-aware AI Planning. Particularly, we focus on an explanatory setting for tasks encoded in a logical formalism, where given an agent model (encoding the task), an explanandum entailed by the agent, and a user vocabulary specifying terms in the task, the goal is to find an explanation that is at an appropriate abstraction level with respect to the user's vocabulary. We propose a logic-based framework aimed at generating such explanations by leveraging a method called *knowledge forgetting*, and present an algorithmic approach for computing them. Our experimental evaluation shows the promise of our framework.

## 1 Introduction

Human-aware AI Planning (HAIP) has been established as a paramount area of research due to its ability to help human users interface with AI agents in complex (sequential) decision-making tasks (Kambhampati 2019). A typical HAIP scenario involves an agent $M_a$ explaining an explanandum $\varphi$ that is inexplicable to a human user $M_h$, where $M_a$ and $M_h$ encode the agent's and the human's version of a planning task (e.g., in a PDDL format). This approach is referred to as the *Model Reconciliation Problem* (MRP) (Chakraborti et al. 2017), and its predominant goal is to provide users with succinct explanations, e.g., explanations of minimal cardinality. However, a common thread around MRP approaches is the assumption that the users understand the task at the same level of abstraction as the agent (Sreedharan, Chakraborti, and Kambhampati 2018; Vasileiou, Previti, and Yeoh 2021). Interestingly, Sreedharan, Srivastava, and Kambhampati (2021) have made some effort toward computing explanations for users at different abstraction levels. Nevertheless, their approach focuses exclusively on state abstractions for specific planning tasks.

In this paper, we center our attention on utilizing the notion of abstract explanations for decision-making tasks that can be encoded using a logical formalism. Particularly, we present a logic-based framework, where given an agent knowledge base $KB_a$, an explanandum $\varphi$ entailed by $KB_a$, and a user vocabulary $\mathcal{V}_h$ consisting of user specified terms,

the goal is to generate an explanation that is at an appropriate abstraction level with respect to $\mathcal{V}_h$. To generate the explanations, we leverage a method called *knowledge forgetting* (Delgrande 2017) and use it to define the notion of *abstract explanations*. We present a simple algorithmic approach for computing abstract explanations, and evaluate its efficacy on a set of SAT-based benchmarks. While the operation of knowledge forgetting has been extensively studied in various settings (Zhang and Zhou 2009; Lutz and Wolter 2011; Wang et al. 2014), its applicability in the context of HAIP has not been explored, to the best of our knowledge.

The motivational drive of this work is rooted in the understanding of Relevance Theory (Wilson and Sperber 2002), which suggests that the relevance of an utterance depends on maximizing the recipient's cognitive effect and minimizing their cognitive effort. In our context, this can be translated as an explanation of minimal cardinality (minimize effort) abstracted enough for a given user vocabulary (maximize effect). This will be the primary objective for generating abstract explanations in this paper.

## 2 Logical Preliminaries

We assume a propositional language $L$ consisting of a finite set of propositional letters $\Gamma$. The simplest formulae in $L$ are *literals*, which are letters or their negations, while more complex formulae can be recursively built up from letters and the classical logical connectives. A *knowledge base $KB$* is a set of formulae. The set of letters used in the formulae of $KB$ is called the vocabulary of $KB$, denoted by $\mathcal{V}_{KB}$. An *interpretation* is a function $\mathcal{I} : \Gamma \rightarrow \{true, false\}$, and if there exists an interpretation that satisfies a $KB$ we say that $KB$ is *satisfiable*, otherwise $KB$ is *unsatisfiable*, denoted by $KB \models \bot$. A $KB$ entails a formula $\varphi$, denoted by $KB \models \varphi$, if and only if $KB \cup \{\neg\varphi\} \models \bot$.

Unless stated otherwise, in what follows we assume that a $KB$ is satisfiable and it is expressed in *conjunctive normal form* (CNF), that is, a conjunction of clauses, each of which is a disjunction of literals.

**Definition 1** (Explanation). *Given $KB \models \varphi$, an explanation for $\varphi$ from $KB$ is a subset $\epsilon \subseteq KB$ s.t. $\epsilon \models \varphi$ and $\forall \epsilon' \subset \epsilon$ we have $\epsilon' \not\models \varphi$.*

**Definition 2** (Minimal Unsatisfiable Set (MUS)). *Given $KB \models \bot$, a subset $\mathcal{U} \subseteq KB$ is an MUS if $\mathcal{U} \models \bot$ and*

$\forall \mathcal{U}' \subset \mathcal{U}, \mathcal{U}'$ *is satisfiable.*

MUSes and explanations are related by the following:

**Proposition 1.** *Given $KB \models \varphi$, $\epsilon \subseteq KB$ is an explanation of $\varphi$ ($\epsilon \models \varphi$) iff $\epsilon \cup \{\neg \varphi\}$ is an MUS of $KB \cup \{\neg \varphi\}$.*

Our framework presented here is closely tied to the foundations laid in Vasileiou, Previti, and Yeoh (2021), namely the logic-based version of the model reconciliation problem (L-MRP).[1] In L-MRP, one is given two knowledge bases $KB_a$ and $KB_h$ of the agent providing an explanation and the human receiving the explanation, respectively, such that $KB_a \models \varphi$ and $KB_h \not\models \varphi$, and the goal is to find an explanation $\epsilon = \langle \epsilon^+, \epsilon^- \rangle$, where $\epsilon^+ \subseteq KB_a$ and $\epsilon^- \subseteq KB_h$, s.t. $(KB_h \cup \epsilon^+) \setminus \epsilon^- \models \varphi$.

Note that an important assumption of L-MRP is that $KB_h$ is known to the agent a priori. Nevertheless, as we will see in Section 4, instead of a full-fledged $KB_h$, we will only require that a user-defined vocabulary $\mathcal{V}_h$ is provided. As this assumption is significantly more reasonable and realistic, we anticipate that our work is a move in the right direction towards practicality.

## 3 Abstractions via Knowledge Forgetting

The notion of *knowledge forgetting*, henceforth forgetting, is taken to be an operation that decreases the language of an agent, insofar as the vocabulary of the agent's language is reduced. Specifically, assume a knowledge base $KB$ over a vocabulary $\mathcal{V}_{KB}$. The operation of forgetting $\lambda \subseteq \mathcal{V}_{KB}$ from $KB$ is the logical consequences of $KB$ expressible over $\mathcal{V}_{KB} \setminus \lambda$. Forgetting is applied in the contents of an agent's knowledge base and is independent of the underlying formalism.

Delgrande (2017) presents a succinct mechanism for computing forgetting for various logics, however, in this paper we focus on its propositional logic treatment.

**Definition 3.** *Let $KB$ be a knowledge base and $\lambda \in \mathcal{V}_{KB}$ a letter in its vocabulary. Define $KB_{\downarrow \lambda} = \{\varphi \in KB \mid \lambda \notin \mathcal{V}_\varphi\}$.*

That is, $KB_{\downarrow \lambda}$ is simply those formulae of $KB$ that do not mention $\lambda$. In the next definition, $Res(KB, \lambda)$ is the set of formulae obtained from $KB$ by carrying out all possible resolutions with respect to letter $\lambda$.

**Definition 4.** *Let $KB$ be a knowledge base and $\lambda \in \mathcal{V}_{KB}$ a letter in its vocabulary. Define $Res(KB, \lambda) = \{\varphi \mid \exists \varphi_1, \varphi_2 \in KB \text{ s.t. } \lambda \in \varphi_1, \neg\lambda \in \varphi_2, \text{ and } \varphi = (\varphi_1 \setminus \{\lambda\}) \cup (\varphi_2 \setminus \{\neg\lambda\})\}$*

Now, Definitions 3 and 4 can be combined to compute forgetting, resulting in the following definition:

**Definition 5.** *Let $KB$ be a knowledge base and $\lambda \in \mathcal{V}_{KB}$ a letter in its vocabulary. Then, forgetting $\lambda$ from $KB$ is defined as $\mathcal{F}(KB, \lambda) = KB_{\downarrow \lambda} \cup Res(KB, \lambda)$.*

Definition 5 can be interpreted as follows: Perform all possible resolutions with respect to the letter to be forgotten,

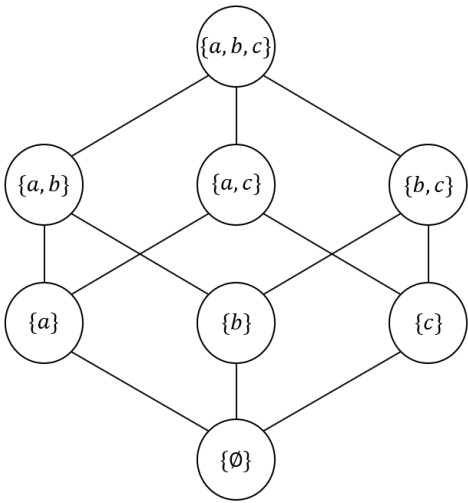

Figure 1: A level-3 abstraction lattice for $KB = \{a, b, \neg a \vee c, \neg b \vee \neg c \vee d\}$. At the root is level-0 of the lattice, i.e., the initial $\mathcal{F}(KB, \{\emptyset\}) = KB$. The child nodes of the root form level-1 of the lattice and represent (from left to right): $\mathcal{F}(KB, \{a\}) = \{b, c, \neg b \vee \neg c \vee d\}, \mathcal{F}(KB, \{b\}) = \{a, \neg a \vee c, \neg c \vee d\}$, and $\mathcal{F}(KB, \{c\}) = \{a, b, \neg a \vee \neg b \vee d\}$. Similarly, the subsequent nodes form level-2, and so on. As one can see, the level of the lattice is specified by the number of letters we are forgetting.

and add these resolvents to those formulae in $KB$ that do not mention that letter.[2] While the resulting $KB$ will be weaker than before, one of the key advantages of this mechanism is that the resulting $KB$ entails the same set of formulae that are irrelevant to what was forgotten. We formalize this in the following corollary:

**Corollary 1.** *Let $KB$ be a knowledge base and $\mathcal{V}_{KB}$ its vocabulary. If $KB \models \varphi$, then $\forall \lambda \in \mathcal{V}_{KB} \setminus \mathcal{V}_\varphi, \mathcal{F}(KB, \lambda) \models \varphi$.*

**Example 1.** *Let $KB = \{a, b, \neg a \vee c, \neg b \vee \neg c \vee d\}$ and $\mathcal{V}_{KB} = \{a, b, c, d\}$, where $KB \models d$, and assume we want to forget $a$. First, we compute $KB_{\downarrow}a = \{b, \neg b \vee \neg c \vee d\}$ and $Res(KB, a) = \{c\}$. Then, $\mathcal{F}(KB, a) = \{b, c, \neg b \vee \neg c \vee d\}$. Notice that $\mathcal{F}(KB, a) \models d$.*

For a more thorough analysis of forgetting and its various theoretical properties, we refer the interested reader to the work by Delgrande (2017).

Importantly now, in this paper we want to utilize the forgetting operation as an abstraction method for a set of formulae. In essence, such an abstraction method will simply be a simplification of the formulae by forgetting a set of letters from their vocabulary. Formally,

**Definition 6** (Abstraction). *Let $KB$ be a knowledge base and $\mathcal{V}_{KB}$ its vocabulary. Then, $\mathcal{F}(KB, \lambda)$ is an abstraction of $KB$, where $\lambda \subseteq \mathcal{V}_{KB}$.*

What is interesting here is that, through the operation of forgetting, we can define an *abstraction lattice* specifying

---

[1]The MRP problem was originally developed by Chakraborti et al. (2017) for (classical) planning tasks.

[2]Note that computing forgetting for a set of letters can be done iteratively, i.e., $\mathcal{F}(KB, \lambda_1 \cup \lambda_2) = \mathcal{F}(\mathcal{F}(KB, \lambda_1), \lambda_2)$.

the abstraction levels that can be achieved on a knowledge base given a set of letters. Figure 1 shows an abstraction lattice based on Example 1.

In the next section, we will take Definition 6 and use it to define the notion of *abstract explanations* with respect to a user-defined vocabulary.

## 4 Explanation Generation Framework

Similar to the concept of MRP, our explanation generation setting concerns an agent explaining an explanandum to a human user. In particular, we assume the following:

- An agent knowledge base $KB_a$ encoding a task (e.g., planning) in a logical language. The agent's knowledge base $KB_a$ is logically closed, insofar as the agent is "logically omniscient" about the problem.[3]

- The user provides the following information to the agent: (i) The explanandum $\varphi$, where $KB_a \models \varphi$; and (ii) a vocabulary $\mathcal{V}_h$ such that $\mathcal{V}_h \subseteq \mathcal{V}_a$, where $\mathcal{V}_a$ is the agent's vocabulary.

As mentioned in the introduction, our motivation for generating abstract explanations is tied to Relevance Theory, where we are interested in finding an explanation $\epsilon$ from $KB_a$ for $\varphi$ that is: (i) cardinality minimal; and (ii) comprises the most letters in $\mathcal{V}_h$. This can be used as an objective function in our explanation generation scheme. To be more specific, (i) suggests minimizing $|\epsilon|$, whereas (ii) can be formalized as maximizing $|\mathcal{V}_\epsilon \cap \mathcal{V}_h|$. As such, the objective function becomes $O(\epsilon, \mathcal{V}_h) = \frac{1}{|\epsilon|} + |\mathcal{V}_\epsilon \cap \mathcal{V}_h|$.[4] An explanation that yields the maximum $O(\epsilon, \mathcal{V}_h)$ will be referred to as the *most-relevant explanation*.

**Definition 7** (Most-Relevant Explanation). *Given $KB_a \models \varphi$, and $\mathcal{V}_h \subseteq \mathcal{V}_{KB}$, $\epsilon \subseteq KB_a$ is a most-relevant explanation iff $\epsilon \models \varphi$ and $\nexists \epsilon' \subseteq KB_a$ s.t. $\epsilon' \models \varphi$ and $O(\epsilon', \mathcal{V}_h) > O(\epsilon, \mathcal{V}_h)$, where $O(\epsilon, \mathcal{V}_h) = \frac{1}{|\epsilon|} + |\mathcal{V}_\epsilon \cap \mathcal{V}_h|$ is the objective function.*

Now, given a most-relevant explanation $\epsilon$ and a user vocabulary $\mathcal{V}_h$, we can compute an abstraction of $\epsilon$ with respect to $\mathcal{V}_h$ using Definition 7, thus yielding the notion of an *abstract explanation*. More formally,

**Definition 8** (Abstract Explanation). *Given $KB_a \models \varphi$ and $\mathcal{V}_h \subseteq \mathcal{V}_{KB}$, $\tilde{\epsilon} = \mathcal{F}(\epsilon, \mathcal{V}_\epsilon \setminus \mathcal{V}_h)$ is an abstract explanation iff $\epsilon$ is a most-relevant explanation.*

In other words, given a most-relevant explanation $\epsilon$, we construct an abstract explanation $\tilde{\epsilon}$ by forgetting the letters that are not contained in the user's vocabulary $\mathcal{V}_h$. Note that we also do not forget the letters in the explanandum $\varphi$ so as to preserve the property that $\tilde{\epsilon} \models \varphi$ (see Corollary 1).

---

---

[3]Albeit a strong assumption, in this paper we suppose that the user is trying to learn from the agent, and as such, the agent encodes the correct information about a specific problem. We plan to relax this in the future work.

[4]Note that one can incorporate weights in the objective function, i.e., $O(\epsilon, \mathcal{V}_h) = w_1 * \frac{1}{|\epsilon|} + w_2 * |\mathcal{V}_\epsilon \cap \mathcal{V}_h|$. This may be of use when one wants to trade-off explanation cardinality and known letters from $\mathcal{V}_h$ (and vice versa).

**Algorithm 1:** Abstract Explanation Generation

**Input:** Agent knowledge base $KB_a$, explanandum $\varphi$, and user vocabulary $\mathcal{V}_h$
**Result:** An *abstract explanation* $\tilde{\epsilon}$ w.r.t. $\mathcal{V}_h$

1   $KB \leftarrow \{\emptyset\}$
2   **for** $c \in KB_a$ **do**
3      $w = |\mathcal{V}_c \cap \mathcal{V}_h|$
4      $KB \leftarrow KB \cup \{(c, w)\}$
5   **end**
6   **while** *true* **do**
7      $\epsilon \leftarrow weightedMUS(KB^{soft} \cup \{\neg\varphi^{hard}\})$
8      **if** $\epsilon \models \varphi$ **then**
9         $\tilde{\epsilon} \leftarrow \mathcal{F}(\epsilon, \mathcal{V}_\epsilon \setminus \mathcal{V}_h)$
10        **return** $\tilde{\epsilon}$
11     **end**
12 **end**

---

**Example 2.** *Let $KB_a = \{a, b, \neg a \vee \neg b \vee c, d, \neg d \vee b\}$ and $\mathcal{V}_h = \{a, d, c\}$, and let $\epsilon_1 = \{a, b, \neg a \vee \neg b \vee c\}$ and $\epsilon_2 = \{a, d, \neg d \vee b, \neg a \vee \neg b \vee c\}$ be two explanations for $c$ from $KB_a$. Further, notice that $\epsilon_2$ is the most-relevant explanation, as $O(\epsilon_2, \mathcal{V}_h) = \frac{1}{4} + 3 > O(\epsilon_1, \mathcal{V}_h) = \frac{1}{3} + 2$. Then, given $\epsilon_2$ and $\mathcal{V}_h$, we get $\tilde{\epsilon} = \mathcal{F}(\epsilon_2, \mathcal{V}_{\epsilon_2} \setminus \mathcal{V}_h) = \{a, d, \neg d \vee \neg a \vee c\}$.*

One of the most important effects of Definition 8 is that, through the machinery of forgetting, we can forge explanations that were not previously contained in $KB_a$. This gives us the opportunity to explore the construction of more personalized and intelligible explanations for human users.

### Computing Abstract Explanations

We now present a simple algorithm for computing abstract explanations. At a high level, the algorithm finds the most-relevant explanation (Definition 7) with respect to the user's vocabulary, and then computes the appropriate abstraction level for that explanation using the forgetting operation. The objective function described in the previous section is incorporated through a weighted MUS procedure that computes the most-relevant explanation by prioritizing the formulae that consist of the most letters with respect to the user vocabulary.

Algorithm 1 presents the pseudocode. The algorithm first weights the formulae in $KB_a$ according to the number of intersections of their letters with those in $\mathcal{V}_h$ and inserts them into the knowledge base $KB$ (Lines 1-4). The main loop starts at Line 5. At Line 6, the algorithm computes a weighted MUS $\epsilon$ on $KB \cup \{\neg\varphi\}$ by treating the formulae in $KB$ as soft and the formula $\varphi$ as hard. We remind the reader that the soft formulae are those formulae that will be removed by the minimizing procedure of weightedMUS, while hard formulae will not. If $\epsilon \models \varphi$, then $\epsilon$ is a most-relevant explanation, and as such, an abstraction of $\epsilon$ is computed (Definition 7) and returned (Lines 7-9). Otherwise, the algorithm continues by computing a new MUS, where already computed MUSes are blocked in order to avoid infinite loops. Finally, the completeness and correctness of Algorithm 1 rests on Proposition 1, as we sketch below.

**Theorem 1.** *Algorithm 1 is complete and correct.*

*Proof.* (Completeness) First, Algorithm 1 always returns a solution, i.e., an explanation from $KB_a$ for $\varphi$. Notice that since $KB_a \models \varphi$, then $KB_a \cup \{\neg\varphi\} \models \bot$, and thus, there exist a set of MUSes $\mathcal{U}$ from $KB_a \cup \{\neg\varphi\}$. From Proposition 1, $\exists \epsilon \in \mathcal{U}$ such that $\epsilon \setminus \{\neg\varphi\} \models \varphi$ is an explanation for $\varphi$ (Definition 1). Since Algorithm 1 computes MUSes from $KB_a \cup \{\neg\varphi\}$ (Line 6), it will eventually find and return an explanation.

(Correctness) Algorithm 1 is guaranteed to return an abstract explanation given $KB_a \models \varphi$ and $\mathcal{V}_h \subseteq \mathcal{V}_a$. This is due to the fact that it uses a $weightedMUS$ function for computing explanations. Firstly, the algorithm creates a weighted knowledge base $KB$ (Lines 2-4), where the weights of the formulae in $KB$ denote how many of their letters are in $\mathcal{V}_h$. The $weightedMUS$ function on Line 6 uses an implicit hitting set process (see the work by Ignatiev et al. (2015) for more) by iteratively building up MUSes from $KB \cup \{\neg\varphi\}$, where its optimization function maximizes the weights and minimizes the cardinality of the computed MUSes. In other words, the optimization function of $weightedMUS$ is akin to our objective function presented in Section 4. This means that the algorithm evaluates candidate explanations $\epsilon$ by prioritizing those according to our objective function. Therefore, if $\epsilon \models \varphi$ evaluates to true, then $\epsilon$ is a most-relevant explanation according to Definition 7, and consequently, an abstract explanation (Definition 8) is guaranteed to be returned (Lines 8-9). □

## 5 Experimental Evaluation

We now present an experimental evaluation of Algortithm 1 on some instances from the SAT competition.[5] We ran the experiments on a Windows machine comprising an AMD Ryzen 7 4.20 GHz processor with 32GB of memory. The time limit was set to 500s. Algorithm 1 was implemented in Python and integrates calls to a weighted MUS oracle through the PySAT toolkit (Ignatiev, Morgado, and Marques-Silva 2018). We used our own implementation for the knowledge forgetting operation. The code will be made publicly available after the peer-review process.

In our experiments, we used the SAT instances as the agent's knowledge base $KB_a$. The explanandum $\varphi$ we used for each instance was a conjunction of backbone literals (e.g., a set of literals entailed by $KB_a$), which we pre-computed using the minibones algorithm (Janota, Lynce, and Marques-Silva 2015). For the user vocabularies $\mathcal{V}_h$, we created four scenarios, in which $\mathcal{V}_h$ was a random selection of $10\%, 30\%, 60\%,$ and $80\%$ of the letters in the agent's vocabulary $\mathcal{V}_a$ (Scenarios 1 to 4, respectively).

Table 1 tabulates the results, where we report the cardinality of $\mathcal{V}_h$, the cardinality of the abstract explanation $|\epsilon|$ returned, and the runtimes of Algorithm 1, referred to as ALG1. We observe that ALG1 performed relatively well and managed to find an abstract explanation in short amount of time. As expected, observe that the size of the explanation

[5] www.satcompetition.org

| Prob. | | Scenario 1 | | | Scenario 2 | | | Scenario 3 | | | Scenario 4 | | |
|---|---|---|---|---|---|---|---|---|---|---|---|---|---|
| | | $|\mathcal{V}_h|$ | $|\epsilon|$ | ALG1 | $|\mathcal{V}_h|$ | $|\epsilon|$ | ALG1 | $|\mathcal{V}_h|$ | $|\epsilon|$ | ALG1 | $|\mathcal{V}_h|$ | $|\epsilon|$ | ALG1 |
| BN | 1 | 79 | 7 | 0.05s | 222 | 11 | 0.06s | 435 | 15 | 0.07s | 578 | 22 | 0.07s |
| | 2 | 77 | 2 | 0.05s | 214 | 13 | 0.5s | 400 | 22 | 1.5s | 500 | 30 | 1.0s |
| | 3 | 94 | 14 | 0.4s | 246 | 30 | 0.8s | 471 | 44 | 2.0s | 624 | 75 | 1.0s |
| SAT GRID | 1 | 48 | 11 | 2.0s | 132 | 18 | 2.0s | 256 | 727 | 2.5s | 341 | 7,049 | 8.0s |
| | 2 | 78 | 2 | 0.05s | 222 | 4 | 0.05s | 434 | 8 | 0.05s | 578 | 18 | 1.0s |
| | 3 | 98 | 2 | 90.0s | 284 | 41 | 89.0s | 563 | 5,040 | 112.0s | 749 | 8,640 | 115.0s |
| COMM | 1 | 1,415 | 31 | 3.0s | 4,189 | 42 | 3.5s | 8,347 | 50 | 3.5s | 11,116 | 51 | 3.0s |
| | 2 | 1,599 | 52 | 4.0s | 4,703 | 54 | 4.0s | 9,357 | 60 | 4.5s | 12,469 | 72 | 4.0s |
| | 3 | 1,427 | 48 | 4.5s | 3,897 | 66 | 5.0s | 9,001 | 57 | 6.0s | 12,431 | 94 | 7.0s |
| LOGISTICS | 1 | 35 | 7 | 0.2s | 99 | 18 | 0.3s | 196 | 26 | 0.4s | 261 | 24 | 0.4s |
| | 2 | 35 | 4 | 0.2s | 99 | 9 | 0.4s | 196 | 21 | 0.5s | 261 | 29 | 0.55s |
| | 3 | 23 | 6 | 0.1s | 62 | 15 | 0.05s | 121 | 29 | 0.1s | 160 | 35 | 0.1s |
| ROVER | 1 | 60 | 7 | 0.3s | 172 | 12 | 0.25s | 339 | 16 | 0.4s | 452 | 16 | 0.45s |
| | 2 | 50 | 12 | 0.45s | 136 | 20 | 0.6s | 267 | 36 | 0.5s | 354 | 48 | 0.5s |
| | 3 | 65 | 16 | 0.5s | 177 | 27 | 0.6s | 343 | 28 | 1.0s | 453 | 37 | 1.5s |
| BMC | 1 | 1,414 | 32 | 3.0s | 4,185 | 37 | 3.0s | 8,432 | 49 | 3.0s | 11,120 | 57 | 3.5s |
| | 2 | 1,600 | 51 | 4.0s | 4,709 | 58 | 3.5s | 9,364 | 61 | 4.0s | 12,414 | 72 | 4.5s |
| | 3 | 1,502 | 40 | 6.0s | 3,832 | 53 | 8.0s | 8,331 | 65 | 6.0s | 11,370 | 90 | 6.0s |
| ACE | 1 | 399 | 5 | 1.0s | 1,194 | 14 | 2.0s | 2,387 | 24 | 2.5s | 3,184 | 28 | 2.0s |
| | 2 | 594 | 11 | 3.0s | 1,771 | 22 | 4.0s | 3,535 | 39 | 5.5s | 4,711 | 43 | 6.0s |
| | 3 | 1,200 | 77 | 2.5s | 3,450 | 77 | 3.0s | 6,824 | 77 | 4.0s | 9,075 | 77 | 4.0s |

Table 1: Evaluation of ALG1 on SAT Instances.

returned depends on the size of the user vocabulary, i.e., the more letters we forget the more abstract the explanation will be. In general, we notice a small trend that suggests that the runtimes of ALG1 increase as the size of the (encoded) knowledge bases and $|\epsilon|$ increase. On the other hand, problem instance 3 of SAT GRID induced the highest runtimes, even for an explanation of cardinality 2 (Scenario 1). This is due to the inherent hardness of finding MUSes, for example, extracting an MUS is in $FP^{\Sigma_P^2}$ (Liberatore 2005). Indeed, part of the performance advantage in ALG1 lies in the effectiveness of the underlying SAT and MUS solvers. This also implies that any advancement in those solvers will automatically reflect in performance gains in our algorithm.

## 6 Discussion and Conclusion

In this paper, we developed a simple framework that is able to generate abstract explanations with respect to a user defined vocabulary. According to Relevance Theory, these explanations can be thought of as a step towards creating more personalized and intelligible explanations for human users, that is, explanations that minimize the user's cognitive effort and maximize their cognitive effect. Importantly, we view this work as a necessary step towards realizing an interactive, multi-shot explanation generation scheme, where human users will be able to interact with an agent in a dialogical fashion. For example, the abstract explanations presented here can serve as the information that instigates the dialogue between the user and the agent. Specifically, we can conceptualize a framework consisting of an agent model $M_a = \langle KB_a, KB_h^a \rangle$, where $KB_h^a$ is an approximation of the user's knowledge (initially empty or filled with domain-specific common knowledge) that is aimed to be updated through the following interactions: Upon receiving the initial abstract explanation from the agent, the user would be able to request further clarification on the explanation by asking for more information, in which case the agent will increase the explanation's granularity, or for less information, in which case the agent will decrease the granularity. This can be achieved by our framework by traversing the abstraction lattice either upwards or downwards (e.g., see Figure 1). Nevertheless, an important consideration here is

what information to reveal or abstract for the user, i.e., what nodes to expand in the lattice. We aim to investigate this endeavor in future work. Now, once the user is satisfied with the explanation, the agent will update the user's approximate knowledge base $KB_h^a$ with this explanation, and as such a more accurate representation of the user's knowledge can be learned, leading to the practical inception of MRP.

To conclude, we proposed a simple logic-based framework that given an agent knowledge base, an explanandum, and a user vocabulary, it generates abstract explanation with respect to the vocabulary. Due to its logic-based nature, our approach has the additional advantage of being able to deal with tasks coming from different settings, so long as the task can be encoded into a logical formalism. In this paper, we showed its utility on propositional encodings.

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
