# OpenReview forum: "On Generating Abstract Explanations via Knowledge Forgetting"
_icaps-conference.org/ICAPS/2022/Workshop/XAIP — XAIP 2022_

### Official Review · Reviewer_d8vJ · 2022-04-26
**An interesting application of knowledge forgetting techniques to the problem of personalized explanation generation.**

**Rating:** 7
**Confidence:** 4

**Review:**

This paper takes a logic-based approach to an important problem, namely that of personalizing explanations to their recipient in terms of the level of abstraction of the provided explanation. The work joins a relatively small body of related work that eschews (at least partially) the assumption that the recipient and provider of an explanation understand some task at the same level of abstraction. In particular, the paper leverages extant work on knowledge forgetting that allows generated explanations to be tailored to the explainee’s vocabulary. Finally, the paper presents evaluation using established SAT benchmarks which demonstrates the feasibility of the approach to this setting of the problem, as well as shows the impact of  the size of the human's vocabulary on the size of the generated abstract explanation.

In sum, the paper will be a nice addition to this year’s XAIP workshop and will hopefully help deepen the discussion surrounding the need for personalized explanations and the different ways to generate them.

-Typos-

Section 5
* in short amount of time. - > in *a* short amount of time.

Section 6
* it generates abstract explanation with respect to the vocabulary. -> generates abstract explanation with respect to the vocabulary. (remove “it”)

-Suggestions and Questions to the Authors-

Although I found the paper generally clear, I have a couple of suggestions to the authors to improve the clarity of the paper and make it more accessible to a reader interested in XAIP.
1. Example 1 serves its purpose thanks to its simplicity and ability to illustrate the various technical concepts discussed in the paper. However, I think readers would benefit from a more concrete planning example that augments Example 1. For instance, the rovers and logistics domains are used in the evaluation, taken from the SAT competition. Perhaps the paper could include an axiomatization of a problem instance of one of these domains and a few sample explanations given to the human to resolve misconceptions about the validity/optimality of a plan.
2. In the evaluation, clarify what generating an explanation means in the planning domains. After reading Vasileiou, Previti, and Yeoh (2021), I believe explanations are likely ensuring that the plan in question is optimal in the human’s model, but it would be good to make this explicit in the paper.
3. Vasileiou, Previti, and Yeoh (2021) discuss in more detail the application of these techniques to XAIP and in particular explain how the user’s model is 'tweaked’ (e.g., by removing n random preconditions from actions in the user’s planning model). Can the framework described in the current paper handle cases where the human’s model is missing not only some precondition of an action, but rather an ‘entire’ action? For example, the human’s model does not contain the pick up action and without it the goal cannot be satisfied. Can an explanation be generated such that the action is added to the human's model? Similarly for cases where the human is not aware of certain parameters of an action.

---

### Meta-Review · Program_Chairs · 2022-04-30

**Recommendation:** Accept
**Confidence:** 5

**Metareview:**

The reviewer notes that the paper looks at a relatively understudied problem within XAIP and presents a reasonable solution to the method. In terms of the next step, the reviewer notes the possibility of looking at cases where an entire action may be missing and also points out a few ways the clarity of the writing could also be improved. All in all, I am confident that the paper would be of interest to XAIP community and would recommend it to be accepted.

---

### Decision · Program_Chairs · 2022-04-30

Accept